# Online Intelligent Perception of Front Blind Area of Vehicles on a Full Bridge Based on Dynamic Configuration Monitoring of Main Girders

**DOI:** 10.3390/s22197342

**Published:** 2022-09-27

**Authors:** Gang Zeng, Danhui Dan, Hua Guan, Yufeng Ying

**Affiliations:** 1School of Civil Engineering, Tongji University, Shanghai 200092, China; 2Zhejiang Zhoushan Sea Crossing Bridge Co., Ltd., Zhoushan 316031, China

**Keywords:** real-time dynamic configurations, front blind area, intelligent online perception, vertical vortex-induced vibration, driving safety, bridge health monitoring

## Abstract

Establishing an online perception mechanism for a driver’s front blind area on a full bridge under vertical vortex-induced vibration (VVIV) is essential for ensuring road safety and traffic control on bridge decks under specific conditions. Based on accelerations of vibration monitoring of the main girders, this paper uses a real-time acceleration integration algorithm to obtain real-time displacements of measurement points; realizes the real-time estimation of the dynamic configurations of a main girder through parametric function fitting; and then can perceive the front blind area for vehicles driving on bridges experiencing VVIV in real time. On this basis, taking a long-span suspension bridge suffering from VVIV as an engineering example, the influence of different driving conditions on the front blind area is examined. Then, the applicability of the intelligent perception technology framework of the front blind area is verified. The results indicate that, during VVIV, the driver’s front blind area changes periodically and the vehicle model has the most significant impact on the front blind area; in contrast, the vehicle’s speed and the times of the vehicle entering the bridge have minimal impact on it. Meanwhile, it is shown that the framework can accurately perceive front blind areas of vehicles driving on the bridge, and identify different vehicle models, speeds and times of vehicle bridge entries in real time.

## 1. Introduction

As the spanning capacity of bridges continues to improve, bridges become softer and more sensitive to wind. Existing engineering has proven that the flutter problem of long-span bridges can be solved by increasing the critical flutter wind speed in the design stage detailing the wind resistance of bridges. However, it is challenging to solve vortex-induced vibration (VIV) through design. Wind-induced vibrations such as VIV are most common in large-span bridges suffering from frequent wind. Self-excitation and self-limitation are its characteristics. VIV has been observed on a variety of bridge types, including suspension bridges (Humen Bridge and Xihoumen Bridge in China, Parrot Island Bridge and Yi Sun-sin Bridge in South Korea, Verrazano Strait Bridge in the United States), cable-stayed bridges (Angel Bridge in South Korea, Severn Bridge in the United Kingdom, Kesock Bridge in the United Kingdom), beam bridges (Volga River Bridge in Russia, Tokyo Bay Bridge in Japan), and arch bridges (Alconetar Bridge in Spain). Although VIV does not lead to the damage of overall structure in a short time, a flutter does, excessive amplitude and acceleration will seriously affect the use function of bridges, and frequent VIV may also cause fatigue damage to components. Therefore, the monitoring, analysis and control of bridges experiencing VIV are essential for traffic safety and bridge maintenance. When a high-order mode VIV occurs on a long-span bridge, the bridge deck will form a convex and concave curve, which will interfere with drivers’ driving sightline and cause drivers’ psychological fear, thus affecting their driving safety. Larsen et al. pointed out that excessive amplitudes of main girders experiencing VVIV would block drivers’ sightline to a certain extent, which not only distracted drivers but also posed a potential threat to the safety of driving on the bridge when evaluating the VVIV performance of the Great Belt East Bridge in Denmark [1].

In addition to the objective factors of road conditions and weather (this article refers to high-order VIV), subjective factors of drivers also affect driving safety. Numerous accidents are caused by the blind area of drivers’ vision [2], and the structural design of the vehicle cab makes it difficult to entirely avoid the blind area of drivers’ vision [3,4]. The blind area of a vehicle is the biggest obstacle to a driver’s perception of the surrounding environment. Furthermore, this may magnify drivers’ immediate psychological response to bridge vibrations and negatively affect their driving comfort, which makes them link sightline disturbance and bridge vibration with the unsafe design of bridges, thereby causing an adverse social impact on the operation of bridges. Almost all vehicle manufacturers have studied the problem of vehicle vision to improve safety performance. The distribution of blind spots in the driver’s field of vision is directly influenced by the design of the vehicle cab, which affects the driver’s driving comfort and safety. Physiologically, some scholars have studied the effects of vehicle speed on the human eye’s visual field. They found that the human eye’s visual field shrunk with speed increasing [5,6,7,8].

Presently, domestic and foreign codes and other related studies are being conducted from the perspectives of vortex vibration limit and the fatigue of structural components caused by vortex vibration [9,10,11,12]. However, less attention is being paid to driving safety. A driver’s sightline is radial when vehicles drive on bridges. The focus of research is mainly on two driving sightlines: one is the tangent line between the driver’s sightline and the main beam where VVIV occurs. The other is the tangent line between the driver’s sightline and the front cover. An extensive amount of research has been conducted on the blind area formed by the undulating main beam for the driving sightline of the former. Accordingly, Chen et al. established a visual blind area model for drivers under bridges suffering from VVIV using the geometric drawing method [13]. Chen and Cao et al. used iterative methods to solve the problem of vortex vibration limits considering driving safety based on this model [14,15]. For the blind area formed by the vehicle’s front cover, driving sightline is under-studied. A study of online perception of the front blind area and the driving safety of an entire bridge deck based on the dynamic configuration monitoring of main girders and their change law is still in progress.

Based on vibration monitoring acceleration data of the main girder, this paper uses a real-time online acceleration integration algorithm to obtain real-time displacements of measuring points, achieves the real-time estimation of dynamic configurations of the main girders through function fitting, and accomplishes the real-time perception of the front blind area for the vehicle driving on the bridge deck experiencing VVIV. It provides a basis for comprehensive real-time evaluation and decision-making regarding main girder traffic during VVIV. Taking a large-span suspension bridge with VVIV as the engineering background, the framework can perceive the front blind spot of vehicles on the bridge as well as different vehicle models, speeds, and times of vehicle entry to the bridge. This can provide a new means of detecting a vehicle’s front blind area, contributing to the intelligent control of bridge deck traffic, enhancing car–road–bridge synergy, and improving smart transportation.

The rest of this paper is organized as follows: Section 2 introduces the basic concept of the front blind area on a bridge and its online perception technology route. Section 3 presents and explains the online perception of front blind area based on acceleration monitoring in detail, including a theoretical derivation of drivers’ visual front blind area, a real-time online acceleration integration algorithm, the fitting of real-time dynamic configurations, and the online perception of front blind area and driving safety. Section 4 contains our application and discussion, namely, the framework of the application for the Humen Bridge, and an applicability discussion of the technical framework under different conditions. Section 5 concludes our proposed online perception framework of front blind area during VVIV, summarizes the main advantages and limitations again, and includes future directions and expected improvements.

## 2. Front Blind Area on Bridge Deck during VVIV: Basic Concept and Perception Technology Route

This section briefly introduces the basic concept of the front blind area on a bridge deck, then briefly explains the change in the front blind area during VVIV; finally, it presents the technology route of the real-time online perception of the front blind area.

### 2.1. Visual Front Blind Area for Driving on the Bridge Deck Experiencing VVIV

The blind area of car driving refers to the area where the driver’s sight is restricted during driving [16]. Obstacles, pedestrians, or vehicles in the blind area of driving can easily cause errors in the driver’s judgment and operation, leading to road traffic accidents [17]. Therefore, the blind area of car driving is regarded as one of the main obstacles for drivers in obtaining road traffic information. As shown in Figure 1, the driver cannot see the blind area in front of a vehicle when in the normal driving position. The front blind area is the blind area of sight caused by the vehicle [18]. The blind area in front of the car is related to the car body’s height, the seat height, the length of the front of the vehicle, the body of the driver, etc. If the distance of the front blind area is not well controlled, it is easy to cause rear-end accidents [17].

The slope of the ground significantly influences the front blind area of the vehicle. When the car is going uphill, the front blind area becomes more extensive; when the vehicle is going downhill, the front blind area becomes smaller, as is shown in Figure 2.

As shown in Figure 3, the front blind area of the driver’s vision changes with the vehicle’s positions and dynamic configurations when the vehicle is driving on the bridge experiencing VVIV. It is the largest at the peak; at the trough, it is the smallest.

### 2.2. Technology Route of Real-Time Online Perception of Front Blind Area

The paper proposes an online perception technology route for the front blind area when driving over a full bridge by monitoring the main girder configurations in real time. The primary contribution is the precise calculation of a driver’s visual front blind area online in real time by using the real-time dynamic configurations of the main beam based on monitoring acceleration data. Then, the online perception of driving safety and the framework’s applicability under different conditions are discussed. The following allows for comprehensive real-time evaluation and decision-making on the main girders experiencing VVIV, and the technical route is shown in Figure 4.

With generality, this paper takes the Humen Bridge as the engineering background to research the above content. The structure of the bridge is a 302 + 888 + 348.5 m twin-tower three-span suspension bridge. The bridge experienced a famous VIV event on 5 May 2020. Although several vibration suppression measures were taken [19], several VIVs still occur from time to time. The owner installed a permanent health-monitoring system on the bridge to keep ahead of the bridge’s VIV status. The components monitored for VIV mainly include acceleration, wind speed, and wind direction. Seven bidirectional (vertical, lateral) acceleration sensors are arranged at eight equal points on the upstream side of the main span: V8–V14 are vertical and H1–H7 are horizontal. Moreover, seven vertical acceleration sensors are placed at the eight equal points on the downstream side of the main span, numbered V1–V7, as shown in Figure 5. Each sensor is sampled synchronously, and the sampling frequency is 50 Hz.

The accelerometer sensors are suitable for the vibration testing of long-span bridges. The sensors have good ultra-low-frequency response performance and high sensitivity [20]. The parameters of the measurement setup of the acceleration sensor are shown in Table 1 [20]. The sampling frequency of the sensors is 50 Hz in the application, and the acceleration signals are transmitted to the signal acquisition and data processor through wired means. Then, data are transmitted to the database in real time, the port of database is accessed through the application program, and the data are analyzed online in real time. The specific situation is shown in Figure 6.

## 3. Online Real-Time Perception of Front Blind Area during VVIV Based on Acceleration Monitoring

Based on the acceleration monitoring information arranged on the main girder of the suspension bridge, this section introduces a technical framework for the real-time online perception of the front blind area during the VVIV of the suspension bridge. The framework requires the real-time acceleration integration algorithm, the function fitting of real-time dynamic configurations, and the theoretical derivation of the driver’s visual front blind area. Then, the online perception of driving safety is briefly discussed.

### 3.1. Theoretical Derivation of Driver’s Visual Front Blind Area

Taking a half-wave of a bridge experiencing VVIV as an example, the simplified vehicle model is a triangle, and the front blind area of the driver’s vision is deduced.

Different vehicles have different h1 and θ. For example, with a car and a bus, the h1 and θ of the car are smaller than those of the bus, but the ratio is larger than that of the bus, so the front blind area of cars is larger. It can be seen from Figure 7 that the front blind area is constantly changing when the vehicle is driving on the bridge suffering from VVIV.

The following describes the calculation process of the driver’s visual front blind area under VVIV. 

Explanation of vehicle position change [21]: the configurations of the main beam suffering from VVIV fluctuate up and down, and the vehicle’s movement from one point to the other point is a curve, which approximates an oblique line when the time step is very small. The oblique line is decomposed in the horizontal and vertical directions, and the horizontal displacement is equal to the vehicle speed multiplied by the time. The ordinate of the vehicle’s position on the main beam at the current moment is determined by the amplitude of VVIV. Therefore, the current coordinates of the vehicle on the main girder are:(1)$$\left\{\begin{array}{c} x_{v}=\dot{u}t\\ z_{v}=v(x_{v},t) \end{array}\right.$$
where $\dot{u}$ represents the vehicle’s speed. $x_{v}$ indicates the abscissa of the vehicle on the main beam at the current moment. $z_{v}$ represents the current ordinate of the vehicle on the main girder. v(x,t) indicates the current configuration expression.

The current slope of the tangent line at the vehicle’s position is:(2)$$\left.\frac{\partial v(x,t)}{\partial x}\right|\begin{array}{c} x=x_{v} \end{array}=\tan\left(\alpha \right)$$
where α is the angle between the tangent line and the horizontal line of the vehicle’s position at the current moment.
(3)$$\alpha =\arctan\left(\left.\frac{\partial v(x,t)}{\partial x}\right|\begin{array}{c} x=x_{v} \end{array}\right)$$

The present coordinates of the eye are:(4)$$\left\{\begin{array}{c} x_{e}=x_{v}+h_{1}\times \sin\left(\alpha \right)\\ z_{e}=z_{v}-h_{1}\times \cos\left(\alpha \right) \end{array}\right.$$
where $h_{1}$ represents the eye height, which is the vertical distance from the eye to the horizontal line.

The equation system is obtained by combining the dynamic configuration equation with the sightline equation at the current moment.
(5)$$\left\{\begin{array}{c} z(x,t)=f\left(x\right)\\ z(x,t)=tan(\alpha -\theta )\times (x-x_{e})+z_{e} \end{array}\right.$$
where $f\left(x\right)$ is the mathematical expression of the current dynamic configuration. θ is the angle between the sightline tangent to the front cover of vehicles and the extension plane of vehicle chassis.

Eliminate z from the system of equations to obtain a one-variable x nonlinear equation.
(6)$$f\left(x\right)-tan(\alpha -\theta )\times (x-x_{e})-z_{e}=0$$

Solve the nonlinear equation to obtain the abscissa $x_{J}$ of the intersection of the sightline and the configuration. The accurate solution of a nonlinear equation is highly dependent on the initial value. By using the solution of the equation at the current moment as the initial value input of the equation in the next frame, the amount of calculation for solving the equation is significantly reduced, and the accuracy of the solution is enhanced.

The length of front blind area $d_{1}$ is:(7)$$d_{1}=x_{J}-x_{v}-x_{wheel-car}\times \cos\left(\alpha \right)$$
where $x_{wheel-car}$ is the distance from the eyes to the vehicle’s front cover.

For the calculation of the front blind area of the driver’s vision, timing starts when the vehicle is on the bridge; the positions of the vehicle on the main beam can be determined by its speed and configuration expression at a given time; from the tangent slope of the configuration at the location of the vehicle, the angle α between the vehicle and the horizontal line can be determined; based on the angle and eye height, the eye’s coordinates are determined; from the angle α and θ, the sightline equation can be derived; integrate the sightline equation with the configuration equation to determine their intersection point; lastly, the horizontal distance from the intersection to the vehicle coordinates is calculated, which is the length of the front blind spot. Due to the vehicle’s front cover having a certain length, the actual front blind area should equal the horizontal distance from the intersection to the vehicle coordinates minus the length of the cover. In fact, the curvature of the road surface should also be considered. The curved road surface should be used as the front blind area, but after comparison, the results are almost the same, so a straight line is used as the length of the front blind area. As a result of the above theoretical derivation, the length of the driver’s front blind area can be determined at any given moment.

### 3.2. Real-Time Online Acceleration Integration Algorithm

Understanding real-time displacements and dynamic configurations of bridges during VVIV is necessary. Therefore, the synchronous multi-point acceleration-monitoring signals are integrated in real time. This paper adopts the real-time acceleration integration algorithm proposed by Zhen et al. [22] which blocks the monitoring acceleration data into sampling frames and calculation frames of a specific length. It then uses the least-squares method to fit the potential baseline to the calculation frames for correction, filters it through a high-pass filter to remove low-frequency noise, and finally integrates it in time domain. The above four steps can be repeated twice to obtain the velocities and displacements of the same sampling frame length, allowing online integration based on acceleration-monitoring data to be accomplished in real time.

A key part of the method is to eliminate noise in acceleration-monitoring signals while reducing manual intervention to a minimum to achieve real-time performance. According to the research of [23,24], the recursive high-pass filter of the following form is selected:(8)$$y_{j}=\frac{1+q}{2}\left(x_{j}-x_{j-1}\right)+qy_{j-1}$$
where $x_{j}$ and $y_{j}\left(j=1,2,3\ldots \right)$ are input and output signals and q is the filter parameter, $q\in \left(0,1\right)$.

The transfer function $H\left(\omega \right)$ of the filter can be written as follows:(9)$$H\left(\omega \right)=\frac{1+q}{2}*\frac{1-e^{-i\omega \Delta t}}{1-qe^{-i\omega \Delta t}}$$
where ω is the filter frequency and Δt is the sampling interval in the sensor equipment of the bridge health monitoring systems, which is a fixed value; e is a natural constant.

To ensure accuracy, the amplitude of $H\left(\omega \right)$ should be as close as possible to 1; usually, $\left|H\left(\omega \right)\right|=0.97\sim 0.99$ is a good choice. The effect of the filter is determined mainly by the parameters ω and q. The filter cut-off frequency of interest $\omega_{c}$ can be used to determine the filter parameter q. When considering suspension bridge structures with long spans, the filter cut-off frequency $f_{c}$ can be determined by the first-order natural frequency $f_{s}$.
(10)$$f_{c}=\alpha f_{s}$$
where α is the filter proportional coefficient, which can be taken as 1⁄5~1⁄3 for long-span bridges. Within this range, the filter is able to completely eliminate low-frequency noise from vibration signals while retaining structural vibration information.

### 3.3. Fitting of Real-Time Dynamic Configuration Based on Monitoring Acceleration Signals

Monitoring data from multiple acceleration sensors arranged in the longitudinal direction of the main beam are synchronously integrated to obtain the dynamic displacements of the main beam at multiple positions in order to obtain its dynamic configurations.

To demonstrate the real-time dynamic configurations of the main girder of Humen Bridge, this paper uses real-time displacement data collected at several points on the main girder, and fits these data points with a proper function. Humen Bridge is a single-span suspension bridge with hinged supports at both ends, so the vertical displacement at both ends of the main girder is 0. When seven acceleration measurement points are integrated and combined with two vertical hinge points, nine displacement measurement points are obtained.

Fitting functions include Fourier series [25], cubic splines [26], sine functions [27], etc. Using the sine function for fitting is based on two reasons: firstly, it is a comparison of fitting effects; secondly, it is a consideration of the mechanism of VIV. The cause of VIV is the Karman vortex street. The fluid vortex and the sin function are separated from the low-pressure area downstream of the fluid, causing the front and rear pressure sinusoidal functions to change, resulting in bridge deck resonance [28,29]. According to the characteristics of VIV, the sine function and Fourier series fitting are selected. Since the sine function fits the original data points without deviating from them, while the Fourier series fits deviate from them, the Fourier series fitting is not performed. The fitting results of the three methods are shown in Figure 8 below.

The fitting formula of the sine function $f\left(x\right)$ is:(11)$$f\left(x\right)=a_{1}*sin(b_{1}*x+c_{1})+a_{2}*sin(b_{2}*x+c_{2})+a_{3}*sin(b_{3}*x+c_{3})$$
where $a_{i},b_{i},c_{i}(i=1,2,3)$ are parameters of the sine function.

Dynamic configurations of the main girders can be obtained by the sine fitting function given by Equation (11). Figure 9 shows the full-bridge configurations sampled for a half period T/2. 

Figure 10 shows the evolution of the dynamic configurations of the girders lasting 4.4 s under VVIV. Compared to the finite element results [19,30], during the VVIV of the Humen Bridge, the vibration mode of the main girder is the second-order vertical bending symmetry, which is the “M” type. The vibration frequency is 0.2268 Hz with the method proposed by Yu et. al [31], and the vibration period is 4.4 s. The maximum displacement of the real-time dynamic configuration of this VVIV is 13.64 cm, and the minimum value is −13.46 cm. The longitudinal dynamic configurations of the main beam are not entirely symmetrical, and there is a certain hysteresis on the right half of the main beam. There are three reasons why this is the case. Firstly, the main beam undergoes VIV according to the natural mode order of a real bridge, and other components (such as masts, cables, etc.) also vibrate in this mode [32]. Secondly, the mode of the bridge is not evenly distributed along the span, and the natural vibration is coupled in different directions in a real bridge [32]. Finally, it may be that the acceleration sensors are not accurately installed on the eight points of the main beam [33].

### 3.4. Online Perception of Front Blind Area and Driving Safety

It is necessary to design an integrated data processing scheme for the streaming acceleration-monitoring data to perceive the front blind area on suspension bridges suffering from VVIV. With this solution, the driving visual field is calculated during VVIV, and the perception of driving safety under VVIV is realized online based on the driving visual field. The data processing scheme consists of the real-time dynamic configurations of the main beams, vehicle locations in real time, and real-time calculations of the front blind area. Figure 11 presents the flow chart of this processing scheme.

Under VVIV, the framework can detect the front blind area of the vehicle, but it is far from sufficient to ensure driving safety. The driver must obey traffic laws, and be able to predict the dangers that may arise in the blind area based on the characteristics of the blind area of their vehicle and drive accurately. These are too challenging for fatigued and elderly drivers [5,7,34], so developing an online data processing system is necessary to ensure driving safety during VVIV. 

As shown in Figure 12, the online assessment of the driving safety of the whole bridge deck during VVIV is planned based on the real-time dynamic configuration monitoring of the main girder.

In summary, the data processing can be divided into four major functional modules, namely, the dynamic configuration module in real time, the module of the relative positions of the vehicle on the main beam, the module of the front blind area under VVIV, and the module for monitoring driving safety in real time. The functions of each module are described as follows:(1)The module for real-time dynamic configurations: This module is used to realize dynamic configurations in real time. The method employs the recursive least squares method to correct the data baseline, filters the low-frequency noise of the monitoring acceleration signal through recursive high-pass filtering, and then synchronously integrates the acceleration signals to obtain the dynamic displacements of the main beam at multiple positions. With displacements serving as the controlled points, the dynamic configurations in real time are generated by function fitting.(2)The module of relative positions of the vehicle on the main beam: Via the vehicle entering the bridge as the starting time, multiply the vehicle speed by the time to determine the abscissa of the vehicle. After placing the abscissa into the dynamic configuration obtained from the previous module, the vehicle’s ordinate on the bridge is obtained.(3)The module of the front blind area under VVIV: From the previous module, the position of the car on the bridge in real time is determined. Then, the angle α between the vehicle and the horizontal line is calculated from the slope of the car position. Next, the slope of the sightline is calculated by subtracting the angle θ. Lastly, from this angle, the eye position is calculated according to the eye height. Following this, the intersection between the sightline and the dynamic configuration is determined, and the front blind spot is calculated.(4)The module for monitoring driving safety in real time: In the environment where the driving safety monitoring system is utilized [34], the detected objects (person, dog, cat, etc.) enter the driver’s blind area. The sensor captures the unique wavelengths of infrared rays emitted by these mammals, and the sensor triggers the alarm module, completing the detection–processing–alarm cycle. The sensor remains silent if no object enters the blind area. Via the real-time information of the acceleration sensors in the bridge health monitoring system, the driver’s front blind area of vision is sensed online. Then, the blind area is transmitted to the intelligent algorithm of the vehicle by wireless communication, such as 5G, to monitor driving safety. Vehicle–bridge synergy can provide a new means of detecting blind spots, integrating traffic into a whole, facilitating traffic management, and assisting the construction of smart cities.

## 4. Application and Discussion

This section discusses the technical framework of the real-time online perception of the front blind area on the real bridge and its technical applicability under different conditions.

### 4.1. Framework for the Application of the Real Bridge

This section uses the real-time monitoring data of the suspension bridge to test the applicability of the front blind area perception technology framework when vehicles drive on the bridge deck. A car is used as the research object, and its two parameters are h1 = 1.2000 m and $\tan\left(\theta \right)=0.1540$. At 80 km/h, the car gets on the bridge when the VVIV wave reaches its highest point, and then timing begins. Based on the monitoring data, the amplitude of VVIV is 14 cm, and the VVIV mode is an “M”-shaped second-order symmetrical vertical bend. The following discussion is based on this mode shape and amplitude, and will not be repeated. Figure 13 shows the online perception of a car’s front blind area at three different moments during driving. 

As shown in Figure 14, the longitudinal displacements of the main beam and the positions of the vehicle constantly change during VVIV, and the driver’s blind area changes along with it. Figure 14 shows that front blind area of the driver’s vision fluctuates slightly. At the maximum peak of the wave, the maximum front blind area is 5.4423 m; 5.4375 m is the minimum front blind area when the vehicle is in the maximum wave trough. When the vehicle is in a wave peak of 0, the front blind area is exactly 5.4398 m. 

Several reasons are responsible for the slight variations in the front blind area: Humen Bridge has a slope of only 1.5%, and the maximum amplitudes of the measured vibrations are about 14 cm. There is a discontinuity in the online perception of the blind area due to the following reasons: data collected by acceleration sensors are discrete, and the changes in the configuration at this moment and the configuration at the next moment are discrete; in addition, data may also be affected by noise.

In the above application in engineering, the technical framework can detect the front blind area in real time online when the large-span suspension bridge is undergoing VVIV. Based on this, the applicability under different conditions can also be discussed.

### 4.2. Discussion of Technical Applicability under Different Conditions

This section discusses the online perception of the front blind area under the conditions of different vehicle types, speeds, and times of vehicle entry to the bridge as examples of how intelligent perception technology can be applied to the blind area of the bridge deck during VVIV. Moreover, the paper makes suggestions on driving safety on bridge decks experiencing VVIV.

**A.** 
**Online perception of front blind area under different models**


This section discusses the online perception of a driver’s front blind area under different vehicle models. The driver’s visual blind area will periodically change due to the periodic movement of the bridge deck.

The driver’s front blind area in a car is larger than that of an ordinary van. The difference in the visibility of the driver’s front blind area in different models is primarily driven by the model and the driver’s visual height. The ratio of eye height and the tangent including the angle of the driver of the car is bigger, so the front blind area of the driver’s vision is larger. Table 2 shows parameters associated with the front blind area of the vehicles’ vision.

Figure 15 shows the perception results of the front blind area for cars and ordinary vans with a speed of 80 km/h when the wave trough is at its largest. In both vehicle models, when the main beam occurs with VVIV, the blind area of the car is larger than that of the ordinary van. The car’s front blind area fluctuates up and down at 54,398 m, while the van’s front blind area is about 21,373 m. The vehicle model has the most significant impact on the front blind area. The two models of vehicles experienced a similar change in their front blind area due to the same bridge experiencing VVIV and the same speed. When the wind–vehicle–bridge intersection is considered, the lower deflection of the main beam caused by the vehicle’s weight further increases the driver’s front blind area.

Different drivers drive the same vehicle, but the eye heights are different; therefore, the visual blind areas differ. When the driver sits comfortably, the eye height is lower, and the blind area is larger. Therefore, the driver should maintain the correct sitting posture so that the blind area can be reduced; otherwise, the blind area will increase, and even some undesirable blind areas will appear.

**B.** 
**Online perception of front blind area at different vehicle speeds**


This section discusses the online perception of front blind areas at different speeds. The car is driven into the bridge when the wave crest is at its largest, and five speeds are considered, namely, 80, 90, 100, 110, and 120 km/h. As a result of the heavy traffic on Humen Bridge during the service period, the standard speed of the main bridge rises from 80 km/h to 100 km/h, and even 120 km/h is projected in the future.

In Figure 16, with different speeds of the car on the bridge experiencing VVIV, the driver’s visual front blind area changes as the car moves. The speeds have less impact on the length of the driver’s blind area than vehicle models. The vehicle’s speed dramatically impacts the driver’s experience time of the visual blind area. The vehicle passes through the whole bridge at different speeds. After about 2.2 s, the blind area is the same again at different speeds, since at these time intervals, it is exactly the half cycle of the vortex vibration, which means that the peak of the wave is 0. At 321 m, 567 m, or near 2/6 or 4/6 of the main beam, the blind area is the same from the perspective of space.

It is essential for a driver to keep a sufficiently safe distance from other vehicles or obstacles and to be aware of the blind area of the vehicle. Research shows that, in a static state, human vision can reach about 210°, but the effective field of vision that can be perceived is only about 70° [3]. The faster the vehicle’s speed, the narrower the field of vision. When the vehicle reaches 100 km/h, the field of vision is about 40°, and only a minimal area can be seen in front of the vehicle. Therefore, when driving on the bridge suffering from VVIV, the vehicle’s speed should be reduced actively to ensure safety.

**C.** 
**Online perception of front blind area at different times of vehicle entry to the bridge**


When a vehicle drives into the main beam experiencing VVIV, the initial vibration amplitude of the main beam has a different impact on the blind area. Initially, the main beam starts vibrating, and the peak of the wave is the largest; then, the peak of the wave is zero, and the trough of the wave is the largest.

The dynamic configuration of the main beam varies with different times of vehicle entry to the bridge, and the online perception of the blind area is also different. Figure 17 shows that when the vehicle is driving over the bridge experiencing VVIV, the front blind area is smaller when the wave trough is from 0 to the maximum, namely, when the vehicle is driving downhill; the front blind area is bigger when the VVIV peak is from 0 to the maximum, namely, when the vehicle is driving uphill. As the vehicle approaches the peak of the vibration wave, the front blind area is the largest. Compared with the moment when the vehicle enters the bridge, the front blind area is the same after about 2.2 s, and the change range of the front blind area is about 4 mm. The changes in the front blind area cannot easily be compared at different times of vehicle bridge entry since the amplitude of VVIV is only 14 cm. 

According to the above discussion, the method proposed in this paper can be used for the online perception of the front blind area of a suspension bridge experiencing VVIV and its perception law. It can determine the front blind area for different vehicle models, speeds and times of vehicles entering the bridge online, proving that the framework is practical and feasible. The perceived front blind area has a periodicity and the period is half the period of VVIV. The vehicle models play a significant role in determining the size of the front blind area. The experience time of drivers’ visual blind area is affected mainly by their vehicles’ speeds. When confronted with the most unfavorable situations—for example, high-order vibration modes, large amplitudes, steep longitudinal slopes, and when the car enters a wave crest at high speed—drivers must be attentive to the change in front blind area. The amplitude of the VVIV monitored in this paper is only 14 cm. As a result, the length of the front blind zone change is relatively small, but the change law is still very periodic. Monitoring a VVIV amplitude of 50 cm will reveal substantial changes, and the regularity of the pattern will be more evident.

## 5. Conclusions

The paper uses the real-time online acceleration integration algorithm to obtain real-time displacements of measuring points based on the monitoring acceleration data of the main girder. It then employs curve fitting to estimate the dynamic configurations in real time. Then, the method for accurately calculating the driver’s front blind area is developed based on the vehicle’s position on the bridge in real time. At last, the framework for the real-time perception of the driver’s front blind area and consequent improved driving safety under VVIV is proposed. 

The front blind area intelligent perception framework can be applied to different vehicle models, speeds, and times of vehicle bridge entry. The specific perception situation is as follows: the front blind area changes periodically, and the period is about half a cycle of the VVIV; the vehicle model is the primary factor affecting the size of the front blind area; the vehicle speed is the primary factor for the driver’s experience time of the blind area. Pay attention to the front blind area to ensure safety during the most unfavorable situations. By using the real-time data of acceleration sensors in the bridge health monitoring system, not only can the framework perceive the driver’s front blind area online, but also it provides a specific application scenario of vehicle, road and bridge collaborative perception for intelligent transportation.

We have also proposed an application. When vehicles enter the bridge, the cameras installed on Humen Bridge can monitor vehicle information, such as vehicle speeds, models, and the moment of vehicles entering the bridge, and transmit this information to base stations installed on the bridge for calculation. Information about drivers’ blind spots and real-time road conditions is obtained, and is then transmitted to vehicles via 5G, radio frequency, Bluetooth, WiFi, other communications means, or via the GPS or GNSS signal receiving equipment installed in vehicles, and information is transmitted to the vehicle control computer system to facilitate the real-time online perception of driving safety.

## Figures and Tables

**Figure 1 sensors-22-07342-f001:**
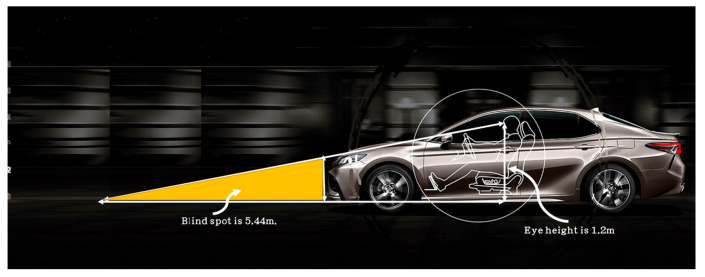
Front blind area of vision when a car is driving on a flat road.

**Figure 2 sensors-22-07342-f002:**
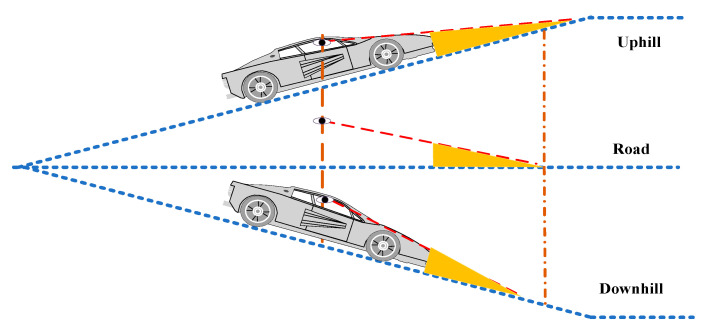
The front blind area of the car changes when the car is going uphill, is on flat road, and going downhill.

**Figure 3 sensors-22-07342-f003:**
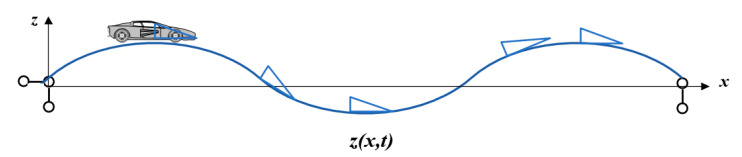
Diagram of the change in the front blind area of the driver’s vision under VVIV (the blue triangle represents simplified vehicle models).

**Figure 4 sensors-22-07342-f004:**
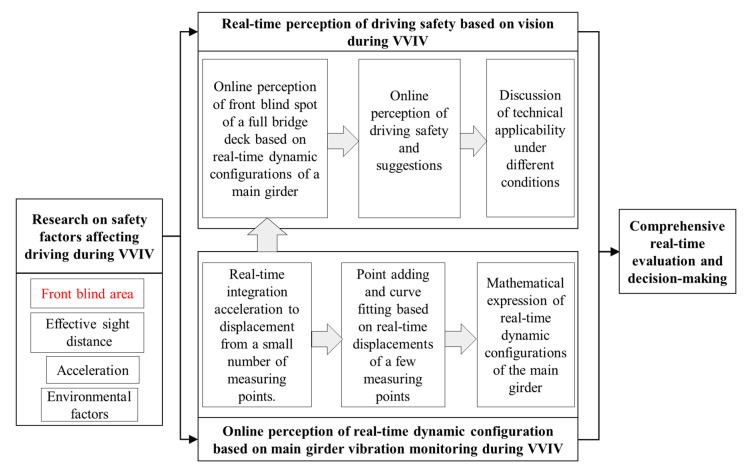
Technical route of online perception of front blind area based on real-time dynamic configurations of main beams.

**Figure 5 sensors-22-07342-f005:**
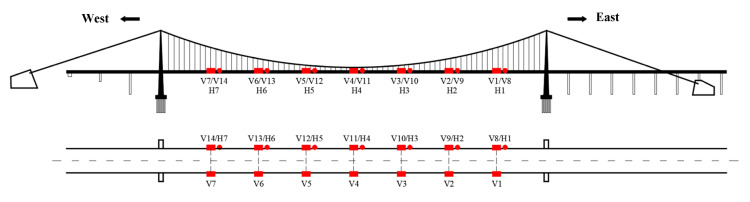
Accelerometer sensor layout on Humen Bridge.

**Figure 6 sensors-22-07342-f006:**
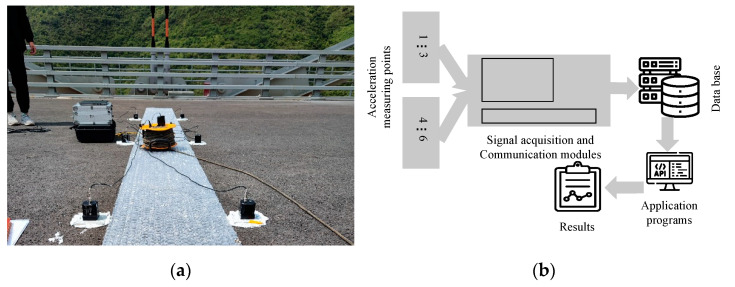
The measurement setup of GT02 force balance accelerometers and diagram of real-time data analysis online. (**a**) The accelerometers in real bridges. (**b**) Diagram of real-time data analysis online.

**Figure 7 sensors-22-07342-f007:**
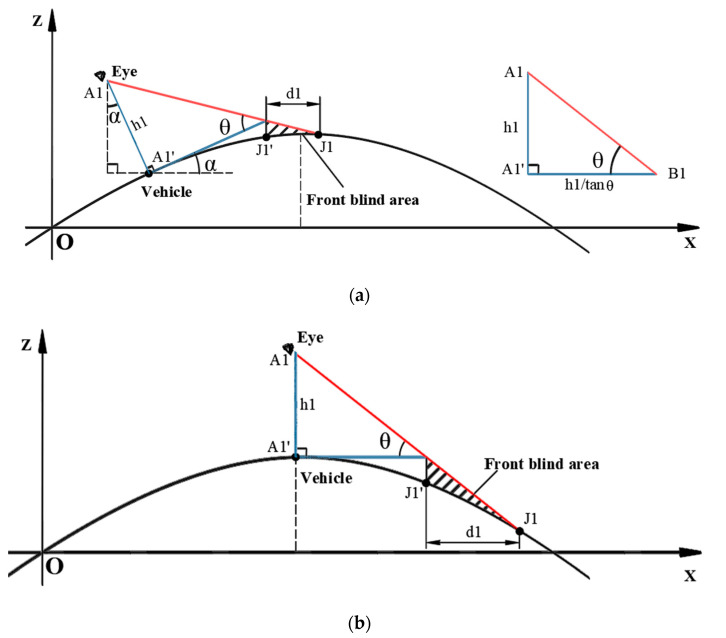
Diagram of front blind area’s calculation. The vertical blue line A1A1′ represents the eye height; the red line A1J1 represents the tangent line between the eyes and the front cover of the car; the black curve represents the current configuration of the main beam; the intersection of the sightline and the configuration is J1; the shaded part represents the front blind area for the driver’s vision; d1 represents the length of the front blind area. (**a**) At time t, when the vehicle drives towards the wave peak, the driver’s visual front blind area is calculated. The triangle at the right of figure is a simplified vehicle model. h1 represents the eye height. θ is the angle between the driver’s sightline which is tangent to the front cover of the car and the horizon. A1 represents the position of the eye. (**b**) At time t + n, the driver’s visual front blind area is calculated when the vehicle is at the wave’s peak.

**Figure 8 sensors-22-07342-f008:**
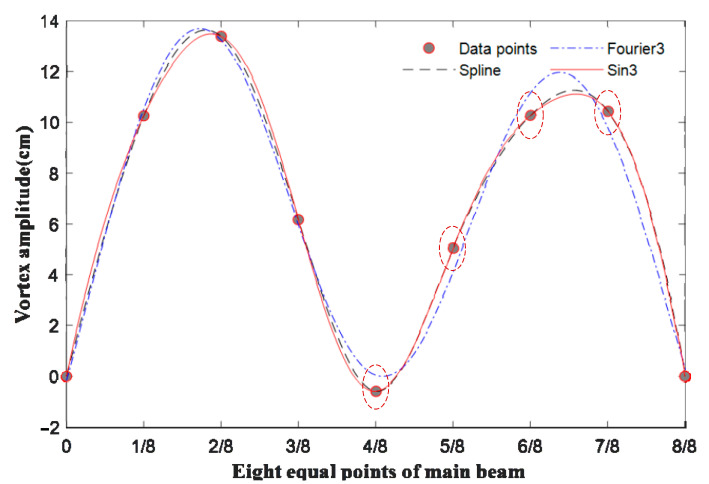
Comparing the fitting effects of the three functions (
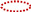
 indicates that the Fourier series fitting function does not pass through control points, but the other two do).

**Figure 9 sensors-22-07342-f009:**
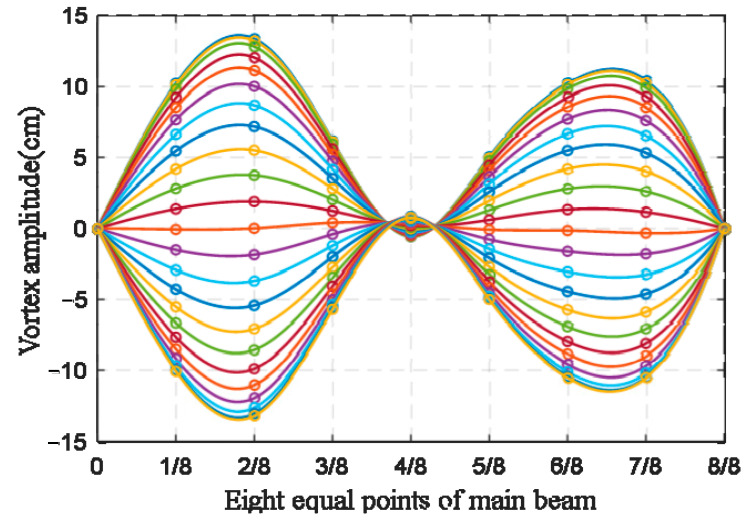
Real-time dynamic configurations of the main beam experiencing VVIV at different times. Each color represents a different moment from the peak of the wave to the trough of the wave.

**Figure 10 sensors-22-07342-f010:**
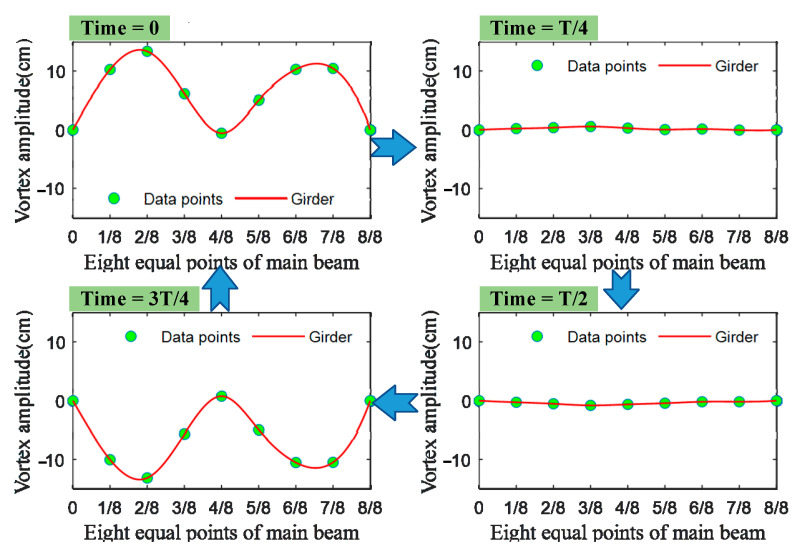
In a single period T of VIV, the longitudinal motion configurations of the bridge (the four times represent the maximum peak, the peak is 0, the trough is the largest, and the trough is 0).

**Figure 11 sensors-22-07342-f011:**
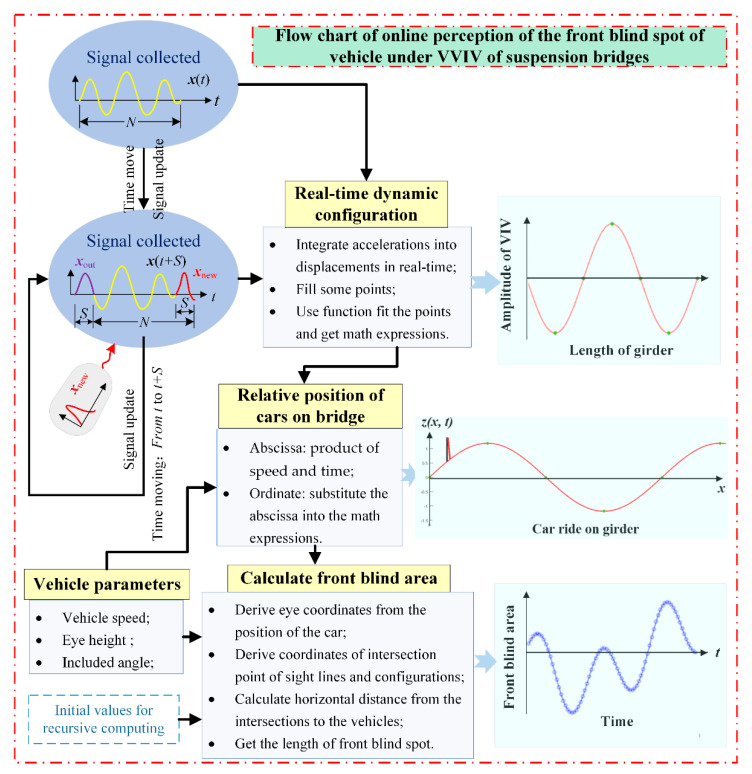
Flow chart of online perception of front blind area of the vehicle on suspension bridge experiencing VVIV.

**Figure 12 sensors-22-07342-f012:**
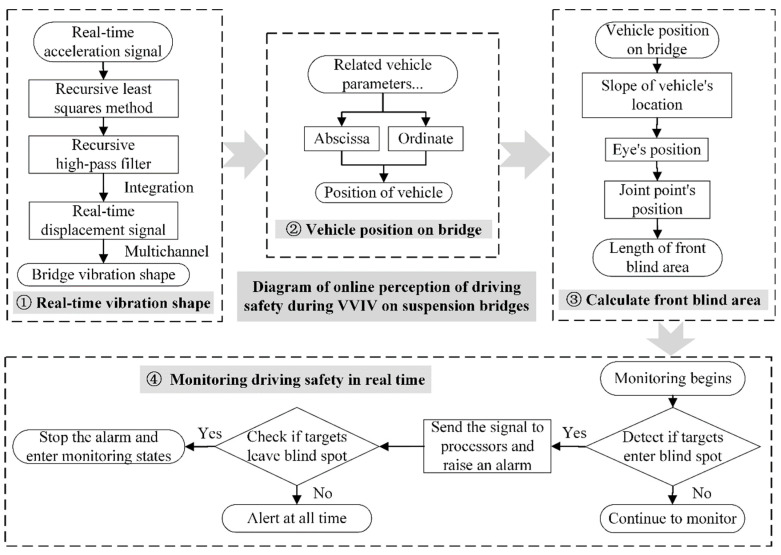
Diagram of online perception of driving safety on suspension bridge experiencing VVIV.

**Figure 13 sensors-22-07342-f013:**
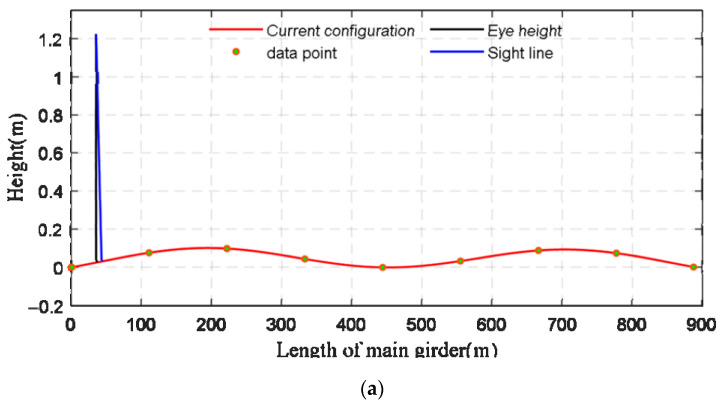
The car’s driving front blind area at three different moments under VVIV (

: dynamic configurations of VVIV;

: the eye height; 

: the tangent line between eye and car’s front cover). Tangent line intersects the configuration at a point, and the length of front blind area is the distance from the intersection point to car’s head. (**a**) t = 1.6 s, online perception of the front blind area. (**b**) t = 16 s, online perception of the front blind area. (**c**) t = 32 s, online perception of the front blind area.

**Figure 14 sensors-22-07342-f014:**
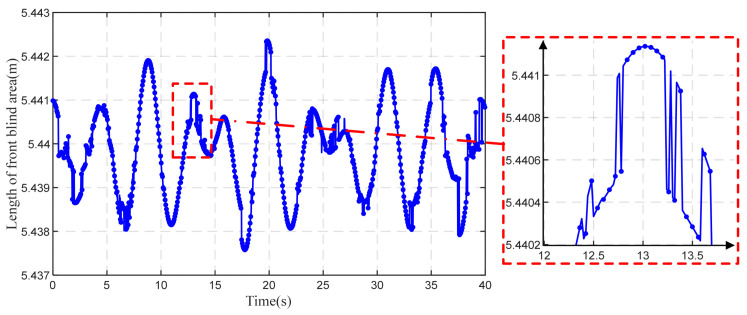
Online perception of front blind area during VVIV.

**Figure 15 sensors-22-07342-f015:**
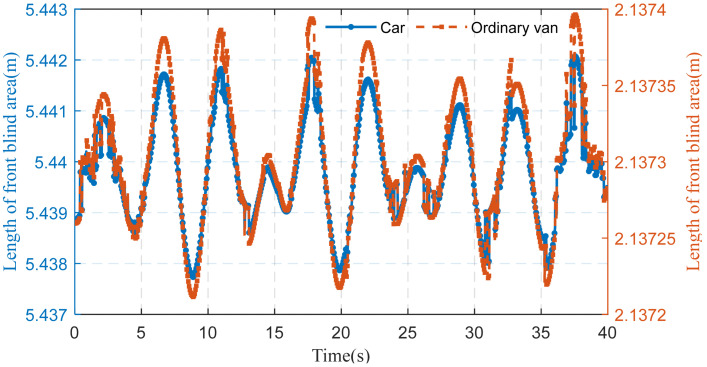
Online perception of front blind area for cars and ordinary vans.

**Figure 16 sensors-22-07342-f016:**
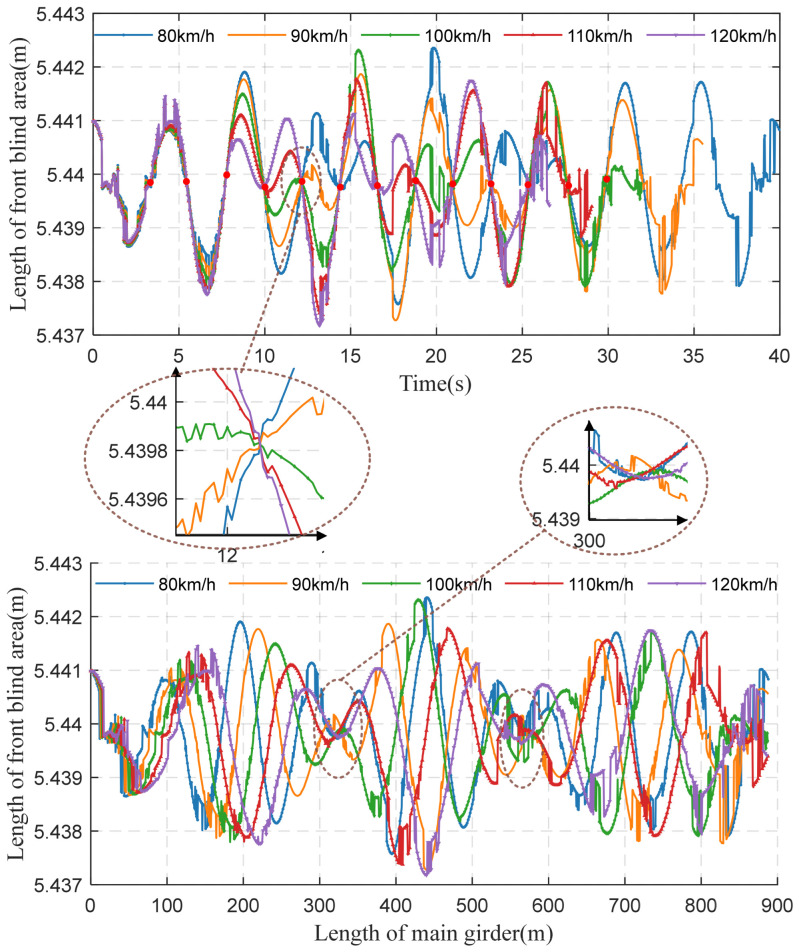
At 80, 90, 100, 110, or 120 km/h, the drivers’ front blind area is perceived online.

**Figure 17 sensors-22-07342-f017:**
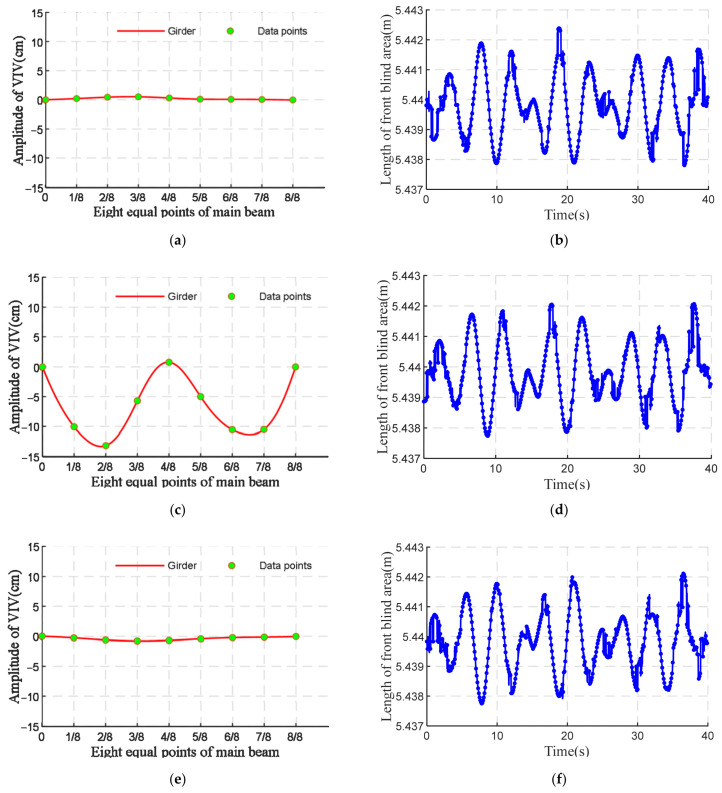
Online perception of front blind area for vehicles entering the bridge at different times. (**a**) When the peaks of the main beam are zero. (**b**) Online perception of front blind area. (**c**) When troughs of beams are the largest. (**d**) Online perception of front blind area. (**e**) When the trough of main girders is zero. (**f**) Online perception of front blind area. (**g**) When peaks of main girders are the largest. (**h**) Online perception of front blind area.

**Table 1 sensors-22-07342-t001:** Parameters of measurement setup of GT02 force balance accelerometer.

Index	Main Technical	Range
1	Measuring range	±2.0g
2	Frequency range	DC−120 Hz
3	Dynamic range	>120 dB
4	Sensitivity	±2.5 V/g
5	Chiasma interference	<3%
6	Linearity	>1%
7	Noise	≤1 ug
8	Temperature drift	≤0.01 %g/°C
9	Zero drift	$<500\times 10^{-6}$ g/°C

**Table 2 sensors-22-07342-t002:** Parameters related to driver’s front blind area.

Vehicle Model	Eye Height (m)	Tangent of Included Angle	Length of Vehicle’s Cover (m)	Length of Front Blind Area (m)
Car	1.2000	0.1500	2.3524	5.4398
Ordinary van	1.4500	0.5000	0.7627	2.1373

## Data Availability

The data are available on request from the authors.

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
