# Peer review of "Online Intelligent Perception of Front Blind Area of Vehicles on a Full Bridge Based on Dynamic Configuration Monitoring of Main Girders"

_sensors, 2022, doi:10.3390/s22197342_

Round 1

Reviewer 1 Report

Dear Authors,

 I have some comments on your article:

1. Literature should be checked if there are no newer items. Especially from the last 18 months.

2. A few more references should be added to the list.

3. Please provide more details in the article about the measurement setup for multiple acceleration sensors.

4. All indexes in symbols in text and equations should be checked carefully.

5. In the Conclusions section, please add information on how to apply the research contained in the article in practice.

Reviewer 2 Report

The authors developed a acceleration based integration algorithm to predict driver’s front blind area on the full bridge, which Real-time online acceleration under vertical vortex-induced vibration could used for road safety and traffic control on bridge decks under certain conditions. It is may useful for helping safety driving but is still questionable.

Firstly, whether the accidents is due to the front blind area? The blind area different is less meter base on calculate of this online system, so how to control the traffic? And also the accuracy is depended the response of sensor, so what sensor used? Could de this system used for auto driving car? This reviewer recommends this paper could be transferred to automation and controlling research based instead of sensors, Thanks.

Reviewer 3 Report

The work presents a good study on the effect of VVIV on the perception of the driver’s front blind area.

Although the results using real data did show that, for this example, the effect of VVIV will not be a huge problem, the basis and how to do future analyses are presented.

Chapters 2 and 3 could have more references, there are many statements, but few references.

Most figures are flawless, but:

Fig. 4 - could have a more flat color palette, even using blank background.

Fig. 7 - the circle must be added to another point.

Fig.10 - the figure does have not enough resolution.

Fig.11 - too much color information and even trace/dot framing.

Round 2

Reviewer 1 Report

Dear Authors,

Thank you very much for introducing changes that have improved the quality of the article. I have no more comments.

Best regards

Reviewer 2 Report

It is a possible approach and valuable to try. For the actual effect still need to be verified since there ismore the environmental noise could be affect this system measurement.   
